# Qualitative analysis to identify determinants of use among different occupational settings and channels of communication to address smokeless tobacco use in Sri Lanka

Hemantha Amarasinghe[1]*, Nilantha Ratnayake[2], Sajeeva Ranaweera[3], Thirupathy Suveendran[4], Palitha Abeyakoon[4,5]

1 Department of Community Dental Health, Faculty of Dental Sciences, University of Sri Jayawardanapura, Nugagoda, Sri Lanka, 2 Provincial Directorate of Health Services, Western Province, Colombo, Sri Lanka, 3 Expert Committee on Tobacco, Alcohol and Illicit Drugs, Sri Lanka Medical Association, Colombo, Sri Lanka, 4 World Health Organization, Country office, Colombo, Sri Lanka, 5 Former Chairman, National Authority on Tobacco and Alcohol, Battaramulla, Sri Lanka

* hemanthaamarasinghe@sjp.ac.lk

**Data Availability Statement:** Field visit notes and script of audio tape recordings are available on

## Abstract

Smokeless tobacco (SLT) use is a leading cause for oral and pharyngeal cancers in the Southeast Asian region which leads to considerable morbidity and mortality. This study aims to Identify the determinants of use and channels of communication to address smokeless tobacco use among specific user groups in Sri Lanka. The study uses a qualitative approach with purposive, snowballing sampling among groups. Specific high-risk demographic and occupational groups that are known to have high prevalence of SLT use were identified in five of the 25 administrative districts of Sri Lanka, were approached. Sixty-two in-depth interviews and 10 focus group discussions were carried out among different occupational groups in five districts. Thematic analysis coding each reported determinants and communication methods was carried out. Users of SLT of different groups revealed different determinants of use. Some of the determinants were common to several groups. When investigated the preferred methods to address SLT, it is found that the media use was also different among these groups. All preferred group level awareness programmes to individual level awareness programmes. Most groups watch specific television channels at specific times of the day. Most groups stated that they accepted the messages of religious leaders and cancer victim groups. Radio and newspapers were used rarely by these groups that were studied. Smart phones were used only by urban youth and others used basic mobile phones only for calling. Different risk groups were identified allowing preparation of an overall communication approach to address use of SLT. The findings here Identify determinants of use and the communication approaches that can be used to prepare an effective communication strategy to address SLT use among different and sometimes hidden groups in resource poor settings in low a middle-income country.

request from the corresponding author. However, these are in Sinhala language and handwritten. Therefore, detail presentation of each visit with photographs were summitted as supplementary file.

**Funding:** This work is supported by the World Health Organization through a grant awarded to PA (FCTC 2030 Project: SESRL 1815768 (65629)). The funder had no role in study design, data collection and analysis, decision to publish, or preparation of the manuscript.

**Competing interests:** The authors have declared that no competing interest exist.

## Introduction

In Sri Lanka, the most widespread smoked form of tobacco is cigarettes, followed by beedi. Chewing and sniffing are the most common methods of using Smokeless Tobacco (SLT). SLT in the form of betel chewing is a deeply rooted lifestyle habit in Sri Lanka especially in the rural areas and estate sector. Betel chewing ingredients such as betel leaf, tobacco, areca-nut and lime are available in the open market or home-grown and no taxes were imposed within the entire supply chain.

Newly introduced commercial preparations of SLT are also becoming popular among SLT users in Sri Lanka. These commercial preparations, namely gutka, pan-parag, hans and mawa are imported from neighbouring countries, while products such as babul and beeda are produced in Sri Lanka [1–3]. However, these newly introduced commercial preparations are not available in the open market and are sold in the grey market, only to known customers.

World Health Organization (WHO) STEPwise approach to chronic disease risk factor surveillance (STEPS) of 2015 showed that 29.4% males and 0.1% females were current smokers in Sri Lanka, while 26% of males and 5% of females current users of SLT [3].

The WHO Global Youth Tobacco Survey (GYTS) 2015 showed that 3.2% of boys and 0.2% of girls between 13–15 years of age smoked at least once during the 30 days preceding the survey in Sri Lanka. The prevalence of SLT use among students of this age-group was 4.2% among boys and 0.5% among girls [4]. A study conducted among those over 30 years of age in the rural areas and in the plantation sector of the Sabaragamuwa Province (one of nine provinces of Sri Lanka) showed that 53.7% of the study population chewed betel daily and 27%of them were ever smokers [5]. Cigar use was very low (1%) among these communities and dual use of smoking and betel quid was 3.2% and betel quid chewing, smoking and alcohol use was remarkably high (20.7%) [5].

Although the SLT import, production and selling are banned in Sri Lanka through a Gazette Notification issued on 1st of September in 2016, under the National Authority on Tobacco and Alcohol Act of 2006, there is no restrictions on use of these substances, except in specific locations such as government institutions. Therefore, there are no legal implications for admitting SLT use.

A large body of published epidemiological studies shows a strong association between SLT use and several serious health outcomes. Among them are cancers of the lip, oral cavity and pharynx and oesophagus [6]. Oral potentially malignant disorders (OPMD), is another condition caused by SLT [5]. Oral cancer is more common in males, which is attributed to high prevalence of tobacco, areca nut and alcohol habits among males. It is the leading cancer site among Sri Lankan males. The incidence of oral cancer among men in Sri Lanka was reported at 19 per 100,000 population in 2019 [7]. Oral cancer accounts for nearly 8.6% of all reported cancers with the highest mortality rates of all cancers in Sri Lanka [7].

Oral cancer usually arises in association with an oral potentially malignant disorder (OPMD), the presence of which substantially increases the risk of developing an oral cancer. The global prevalence of any OPMD is estimated to be a mean of 4.47% [8]. However, a higher prevalence is reported from South and South East Asia e.g. Taiwan 12.7% [9], Papua New Guinea 11.7% [10]. In Sri Lanka, the prevalence of oral leukoplakia (OL) and oral submucous fibrosis (OSF) was reported as 2% and 1.5% [11].

Studies show that public awareness of the harms of SLT use, OPMD and its risk factors were low [12, 13]. Moreover, only 24% of the population aware that areca nut is carcinogenic [12].

Effective prevention of SLT use requires effective communication, based on the determinates of use. These should be culture and context sensitive to be acceptable and impactful.

Success of a communication campaign depends on the message, type of tool, access, and choice of the media by the high-risk group. This study aims to Identify the determinants of use and channels of communication to address smokeless tobacco use among specific user groups in Sri Lanka.

## Methods

### Study design and settings

This was an exploratory, qualitative study to inform effective anti-tobacco messaging by gathering data on reasons for SLT use and preferred communication methods among various groups. Data was gathered through in-depth interviews and focus group discussions (FGDs) from selected user groups and groups at risk. The study was conducted in Puttalam, Colombo, Kurunagala, Kegalle and Ratnapura, five of the 25 districts of Sri Lanka. The districts were selected based on the spread of the different occupational groups identified for the study. We followed the Consolidated Criteria for Reporting Qualitative Research (COREQ) checklist for reporting.

**Study group and the sample size.** Purposive and snowballing sampling techniques were adapted for the study by picking different types of participants who regularly use SLT. Sampling distribution pattern is presented in Table 1. These occupational groups were selected based on the results of the previous qualitative studies conducted in Sri Lanka [2].

Sixty-two in-depth interviews and 10 FGDs were carried out in five districts in the 18–60 years age group during the period of 1st May to 30th September 2019. Participants of the interviews were selected to achieve maximum representation of the socio-cultural and geographic diversity of SLT usage. Investigators visited their occupational settings. Those who had engaged in occupation for more than 6 months were included for data collection.

Interviews were carried out by four interviewers holding a post graduate qualification in Dental Public Health. Two of them were Specialists in dental public health having more than 10 years of experience who had conducted qualitative research in the past. Other two investigators were Trainees in Dental Public Health, who were trained for this purpose by a Specialists in Dental Public Health. The interviews were carried out under the supervision of the specialists. The interviewer guide was prepared, and the areas of focus was developed by a panel of public health experts.

**Table 1. Distribution of data collection sites and engaged communities.**

| Group | District | In-depth interviews | FGDs |
|---|---|---|---|
| Fishing community | Chilaw | 07 | 01 |
| Plantation farming workers | Rathnapura | 06 | 01 |
| Gem mine workers | Rathnapura | 05 | 01 |
| Trishaw taxi drivers | Colombo | 05 | 01 |
| Urban youth | Chilaw | 07 | 01 |
| Construction site workers | Colombo | 05 | 01 |
| Garbage collectors of the local civil administration | Kurunegala | 04 | 02 |
| Bus drivers and conductors | Colombo | 05 | 01 |
| Private security officers | Kurunegala | 04 | - |
| Paddy farmers | Kegalle | 03 | 01 |
| Public state clerks | Kurunegala | 03 | - |
| Self-employed | Kegalle | 02 | - |
| Muslim community | Kegalle | 05 | - |
| **Total** | | **62** | **10** |

Administrative clearance was obtained through the Provincial and Regional Directors of Health Services of the Ministry of Health of the respective districts. Five Medical Officer of Health (MOH) areas were randomly selected from each district, and initial contacts of the high-risk individuals and groups were made with the help of Public Health Inspectors (PHIs) of the MOH area. On average, there are around 15 MOH areas in a district, depending on the population.

All information given by the respondents was written in a record sheet and also recorded using an audio recorder. Each interview required 1 ½ to 2 hrs. Field notes were prepared and interviews were conducted in each group until saturation levels were achieved.

Based on the information gathered from the interviews, ten FGDs were conducted among nine different groups. The objectives of these FGDs were to ensure credibility of information gathered by in-depth interviews, clarification of contradictory ideas and views of the respondents and to identify views on possible preventive strategies. Each focus group consisted of 6 to 10 respondents of similar demographic or occupational groups and required approximately 2 hours to complete. The discussions were conducted by the specialist in Dental Public Health using a standard guideline and was recorded using an audio recorder.

## Data analysis

Thematic analysis was carried out on the qualitative data that was gathered from interviews and FGD.

Two Trainees assigned preliminary codes to describe the results of both in-depth interviews and FGDs. Careful review of the preliminary codes was carried out by two Specialists in Dental Public Health, to ascertain the themes of determinants and communications methods. To enhance the reliability of the data, ten percent of the sample were double coded by both Trainees.

## Ethics statement

Ethical clearance for the study was obtained from the Ethical Review Committee, Faculty of Medicine, University of Colombo, Sri Lanka (EC-19-073). Permission from all relevant parties was obtained. All interviewers were given an identity card to ensure their safety during data collecting on the field.

Informed written consent was obtained from all respondents of in-depth interviews and participants of FDGs. Study subjects who were using tobacco was offered an awareness session and brief interventions to quit the habits following the interview. They were followed up according to the tobacco cessation guideline prepared by Faculty of Dental Sciences, University of Peradeniya, Sri Lanka.

## Patient and public involvement statement

Patients and the public were not involved in the study design or in conducting the study. Results of this study was summarized at the 'National summit on Tobacco control' commemoration on 5th December 2019. Summery findings which contain views of study participants were reported to the National Cancer Control programme, Ministry of Health for development of future communication tools.

Workshops were conducted and disseminated the findings of the study to the public health staff at the grass root level who are closely working with the occupational groups that were studied.

## Results

Sixty-two in-depth interviews and 10 FGD were carried out in above five districts. Table 1 shows the distribution of data collecting sites and the engaged communities. Females were represented among plantation farming workers, garbage collectors of the local civil administration and Muslim communities and income remain constant for each occupation. Non-participation rate was zero due to involvement of the Public Health Inspectors who work closely with these groups in organising data collection.

### Fishing community

Seven in-depth interviews and one FGD were carried out in those who are engaging fishing as an occupation. Illiteracy is common in this community. This group mainly use betel quid during fishing in the sea. The main reasons given for use was to face the cold weather and wetness. The other reported determinants were to prevent feeling lonely due to being away from home, prevent sleepiness and the enjoyment of reflex of chewing. They watch a few specific television channels when on shore (Hiru, Derana, Sirasa) but do not listen to the radio or read newspapers. They prefer to see posters or billboard displays at the corner of the street of the fishing village which should not contains words or sentences but only pictures and graphics. They suggest restricting the availability of SLT products, such as mawa, barbul, beeda and hans, and respect advise by Catholic priests or cancer victims. It was suggested that regular awareness programmes be conducted by the local health staff during regular fisheries society meetings.

### Plantation farming workers

Six in-depth interviews and a FGD were carried out with this group. This group chews betel quid to avoid the cold and wetness felt due to constant rain and the cool climate, to break the monotony while working alone and to maintain company with others. Tea plantation farming workers use betel quid to remove the leeches that sometimes attach themselves on the workers' legs. Rubber plantation farming workers also chew betel to avoid bad odours from the interim products of rubber processing. The Estate worker group was also mostly illiterate and watched specific TV channels (Shakthi and Sirasa) for information. They too did not listen to radio. They suggested that messages to be incorporated into television dramas, home visit education, poster displays at public places and education at society meetings, restrictions of the sale of SLT at worksites and advise from religious leaders to control betel chewing.

### Gem mine workers

Five in-depth interviews and a FGD were carried out with the workers engaged in gem mining. This group chewed betel quid to avoid wetness, coldness and to tolerate the smell of mud inside the mines. They claimed that when chewing betel quid, the body automatically becomes warm when mining in deep grounds. They avoid smoking as it causes difficulties in breathing inside the mines. They avoid alcohol because they believe mining should not be carried out under the influence of alcohol. Other reported determinants were loneliness due to being away from home and the product being available at worksites. They watch television, especially dramas. They do not use smart phones. Most of them were regular betel chewers and suggested to impose regulations, block the supply chain and impose fines to sellers. They also suggested to conduct awareness programmes at the gem mines and at religious places.

### Transport workers: Trishaw taxi drivers, bus drivers and bus conductors

Ten in-depth interviews and two FGDs were conducted with the trishaw taxi drivers and bus drivers and conductors. The main reasons given for use of SLT were to avoid sleepiness and hunger. Avoiding laziness and monotony while waiting for long periods until they get the turn to run the bus was other reason given. They also use betel quid chewing as a mouth freshener. Both groups watch specific television channels (Sirasa and Derana) after 8pm. The do not use smart phones but use ordinary mobile phone only for telephone calls. They too suggested enforcing law and restricting the supply of SLT to control use. They felt stickers carrying messages inside the three wheelers and buses will be effective and also proposed developing live television programmes where cancer victims speak. Bus drivers and conductors proposed to develop short video clips for distribution in CDs or pen drives which carry health messages to be displayed on the televisions installed in buses. Display of health messages in the public notice boards and billboards was their other suggestion. Bus drivers were of the opinion that providing facilities for professional cleaning and polishing teeth through mobile health facilities will help them to quit the betel quid chewing habit.

### Urban youth

Seven in-depth interviews were conducted among 7 school leavers in Colombo district and one Focus Group Discussion was performed for young youth from Puttalam district. The main reasons given for use by this group were peer pressure and to obtain energy to participate in sporting activities. Most stated that they first started use at funeral house gatherings where it is usually freely available.

Urban youth were different from the other groups discussed so far. They live in the Information Technology era, use smart phones, and apps such as face-book and Instagram. Most of them watch YouTube regularly. They watch movies and television films and watch specific channels (Hiru, Derana, Sirasa, Soorian, Shakthi) and respond to text messages. They suggest health messages could be attractive if given through popular female televisions stars. They do not think that posters and leaflets are effective.

### Construction site workers

Five in-depth interviews and one FGD were carried out with the construction site workers. Construction workers stated that they chew betel quid to obtain energy to work throughout the day and to concentrate on the work. They also stated that it is a common practice at the worksite and are induced by seeing others chewing. They watch specific television channels (Derana and Hiru) around 7.30pm, do not use smart phones and have no access to newspapers. They believe that health education material should not contain disgusting pictures like large cancers/tumours. The possible way to convey any messages to them was through the safety-officer's meetings held at the work site every morning, before starting work. They insist on enforcing law to combat SLT and suggested to display posters at notice boards of worksites.

### Garbage collectors of the local civil administration

Four in-depth interviews and two FGDs were conducted with this groups in two different districts. The main reason they used SLT was to tolerate the unpleasant odour at the site of garbage collection. They also do not feel hunger or feel tired when chewing betel quid. They were mostly illiterate, watch TV, mainly news (Shakthi, Sirasa, Hiru and Derana) and do not use smart phones, nor listen to radio or read newspapers. They suggest TV advertisements during news broadcasts, also said that they would trust doctors or cancer victim's messages. They also

suggested health education through home visits and conducting awareness programmes at Buddhist temples and Catholic churches.

### Private security officers and public state clerks

Four in-depth interviews with Private security officers and three in-depth interviews with public state clerks were carried out. Private security officers were chewing betel quid to counteract sleepiness, tiredness, and loneliness. They expressed that they become relaxed and enjoy the chewing reflex. Both groups watch television (Hiru and Derana) during the news time. They also listened to the radio during the night duty time and also read newspapers. There was no smart phones use. Mobile phones were used only for telephone calls. The Private security officers proposed to incorporate health messages in newspapers and radio programmes at night. Both groups encouraged workplace health promotion activities and advise by religious leaders.

### Paddy farmers

Three in-depth interviews and one FGD were conducted. Paddy farmers chew betel quid to feel energetic and feel warm during working. They also keep betel quid to stop attacks by water leeches. All the ingredients for the quid are available at the home garden except tobacco, which is bought from the nearest shop. Paddy farmers watched specific television channels (Derana, Hiru and Suwarnawahini) during lunch and evening news times and watched television dramas at night and read Sunday newspapers. They proposed poster campaigns in market areas, stories from victims and short advertisements between teledramas to address SLT use.

### Self-employed small shop owners

Two in-depth interviews were carried out in two districts. The main reasons given for use was to prevent unpleasant odour in the mouth, maintain the company with others and to avoid sleepiness during working were the main reasons for their use. They watched specific television channels (Hiru, Derana) and read newspapers at their shops. They also had informal conversations with customers who visit their shops. Some had seen anti-tobacco messages when they visited the hospital for treatment. Communicating through religious leaders was suggested as a suitable strategy.

### Muslim community

Five in-depth interviews were carried out in a Muslim community (which is a minority community in Sri Lanka) in a rural area. Snuff use is the main practice among them and main reported determinant of use of snuff was to avoid sleepiness. Young people of this community used social media such as Facebook and Instagram. Women watched specific television channels (Derana and Suwarnawahini) during lunch and night-time and listened to specific radio channels (Vasantham and Hiru). This group felt that short videos by victims would be effective. They also wanted the strict implementation of the ban of SLT and specific progrmmes for their community. Awareness programmes after Mosque services on Friday and speeches during welfare society meetings were recommended as the best ways to reach them.

### Results summary

The reported determinants of use for different groups are summarized in Table 2.

Table 3 identifies the communication channels that should be used for different groups with the best times for communications, with the determinants that should be addressed at different times through different communication methods.

**Table 2. Proximal reported determinants of use.**

| Reported Determinant | Groups |
|---|---|
| Sleepiness | Fishing community, gem mine workers, trishaw taxi drivers, bus drivers and conductors, garbage collectors of the local civil administration, Private security officers, self-employed persons, Muslim community. |
| Energizer, improve efficiency, reduce laziness, reduce tiredness, improve alertness, improve concentration | Urban youth, trishaw taxi drivers, construction site workers, bus drivers and conductors, garbage collectors of the local civil administration, Private security officers, Paddy farmers |
| Loneliness | Fishing community, gem mine workers, plantation farming workers, construction site workers, bus drivers and conductors, garbage collectors of the local civil administration, Public state clerks, Paddy farmers |
| Peer pressure / Induced by peers / to maintain company with others | Urban youth, plantation farming workers, gem mine workers, construction site workers, Paddy farmers, self-employed persons |
| Enjoy reflex of chewing | Fishing community, bus drivers and conductors, garbage collectors of the local civil administration, Private security officers, Paddy farmers |
| Counteract unpleasant odour / taste / feel freshness in mouth | Plantation farming workers (rubber), Gem mine workers (smell of mud), garbage collectors of the local civil administration (smell of garbage), Public state clerks (unpleasant taste after meals). |
| Wetness / dampness / cold | Fishing community, state workers, gem mine workers |
| Availability | Urban youth, gem mine workers, Paddy farmers |
| Keep away leaches | Plantation farming workers, Paddy farmers |
| Break monotony of work | Plantation farming workers, bus drivers and conductors |
| Induce sleep | Urban youth |
| Induce passing stools | Plantation farming workers |
| Relax | Private security officers |
| Quit smoking | Public state clerks |
| No other entertainment / leisure | Gem mine workers |

## Discussion

The different groups surveyed used different media at different times. The reported determinants of use and the beliefs among the different groups were different although there were some overlaps (Table 1). The reported determinants will be useful in delivering tobacco cessation interventions to the specific groups. Primary health care staff and the dental/medical professionals involved in brief interventions and tobacco counselling should be made aware of the results of this study on determinants. These findings clearly demonstrate that when communications on preventing smokeless tobacco is planned, it needs to be carried out after careful consideration of results of this study on determinants of use, methods and channels of communication, to obtain effective outcomes.

In Sri Lanka, overall literacy rate of those aged 10 and over was 95.7% [14], but most of the high risk groups except urban youth were illiterate or only partially literate. Therefore, many groups such as the fisheries community requested to use only pictures in posters to convey messages without using words.

Most of the high-risk groups prefer watching specific television channels at specific times. Therefore, careful consideration of the content and the target groups are necessary when a mode of communication such as television is used. Development and telecasting advertising is considered as expensive but, it has shown to be cost effectiveness in the literature [15].

**Table 3. Communication strategy.**

| Communication Methods | Group | Determinants to be addressed |
|---|---|---|
| TV Channel s: Hiru, Derana, Sirasa, Day time | Fishing community | Wetness, cold weather, dampness, sleepiness |
| TV Channels: Hiru, Derana, Sirasa 6-9pm | Gem mine workers | Wetness and cold weather, unpleasant smell of mud |
| TV Channels: Derana, Hiru After 8 pm | Bus and trishaw taxi drivers | Avoid sleepiness and energize |
| TV Channels: Derana, Sirasa After 7.30pm | Construction site workers | Improve concentration and energize |
| TV Channels: Sirasa, Hiru, Derana, Shakthi News time | Garbage collectors of the local civil administration | Unpleasant feeling, don't feel tired, enjoy chewing reflex |
| TV Channels: Hiru, Derana, Swarnavahini Lunch, evening and night time news and teledrama time | Paddy farmers | Energized, wet and cold feeling, loneliness, enjoy chewing reflex, delay main meals and prevent bad breath |
| TV Channels: Shakthi, Sirasa 6-9pm | Plantation farming workers | Break monotony, wetness and dampness |
| Radio- Hiru, NethFM, YFM, Shaa, Derana, Sooriyan, Shakthi | Urban youth | Peer pressure, to energize |
| Smart phones: Facebook, Instagram | Urban youth | Peer pressure, energize |
| Text messages | Urban youth | Peer pressure, energize |
| YouTube | Urban youth | Peer pressure, to get energize |
| Radio Night time | Private security officers | Sleepiness, tiredness, loneliness, relax and enjoy chewing reflex |
| Posters with pictures and photos | Bus drivers and conductors | Avoid sleepiness and energize |
| Stickers in buses, Three wheelers | Bus drivers and conductors | Avoid sleepiness and energize |
| Newspapers: Sunday Lankadeepa | Famers | Energized, wet and cold feeling, loneliness, enjoy chewing reflex, delay main meals and prevent bad breath |
| Leaflets | None | No use |
| Telephone | None | No use |
| Display poster at hospital | Self employed | Unpleasant feeling, sleepiness, maintain company with others |
| Religious Leaders | Preferred method by 8 high risk groups: Plantation farming workers, gem miners, MC labourers, Private security officers, Public state clerks, self-employed, Muslim communities | Address the groups and determinants as above |

Although newspaper and radio advertising are important components of mass media campaigns, our study found that only 4 groups listened to radio and reads newspapers. Some groups believed that newspapers, leaflets and posters are not effective in delivering messages related to use of smokeless tobacco.

Another significant finding was that most of our high-risk groups use mobile phones only to make telephone calls. Smart phone use and social media use was only seen among the urban youth. Therefore, social media campaigns will reach only this group, and should be tailored to only to this group. More general approaches such as billboards and posters could be used to convey the messages to several risk-groups, but more prominence need to be given to the graphics rather than the words. Many risk-groups agreed that messages from victims would be a good approach. This has also been shown in India [16].

Use of testimonials of tobacco victims has been identified as an innovative communicative approach for smokeless tobacco control in India which could be easily adapted to any low and middle-income country with high incidence of SLT related cancers [17]. Development of social marketing tools and sustaining a media campaign for at least 3 month is very costly for a low and middle-income country. However, in India it has been shown that mass media campaigns can be cost effective and economically justified on low-income and middle-income countries [17].

A total ban on selling, production and import of SLT was imposed in 2016 in Sri Lanka through a Regulation under the National Authority on Tobacco and Alcohol Act of 2006 [18]. This ban was not enforced on betel quid due to socio-cultural issues. However, all the high-risk groups believe that ban should be implemented to restrict the availability to prevent and reduce use. Imposing restriction of availability of traditional betel quid ingredients could be difficult as those ingredients are homegrown in rural areas.

Religious leaders play a pivotal role in changing of behaviour among their devotes. This study showed how several groups could be approached by religious leaders. In 2012, the National Cancer Control Programme of Sri Lanka introduced a new concept for Buddhists to change the ingredients of betel tray (omitting tobacco,areca nut and lime) which is used to invite priests for any religious activity [19]. Approaching religious leaders to partner in the communication campaign is therefore an important component.

However, it must be stressed that great care is required for planning the content of the communication. For example, if a group used SLT to address loneliness or to prevent a bad odour while working, alternative could be introduced. All the above reported determinants should be therefore carefully considered and effective contents of interventions to address each such determinant should be identified. Implementation authorities could identify and prioritise interventions that are common to different occupational group. Unless this is done rigorously, the outcomes would not be what is expected, although the proper communication channels and strategies are used. The most effective approach to developing such contents should be through discussions with the specific groups themselves. They can be involved in developing the material, which can be pilot tested in similar groups elsewhere. Further qualitative studies in different settings are needed to confirm the determinants and methods of communications to curtail the use of SLT.

## Conclusion

Since addressing smokeless tobacco use has to become a high priority public health issue in Sri Lanka as well as in other South Asian countries considering its prevalence of use and harm it causes, scientific but context and group specific interventions will be required to address this issue in the medium and long term. This qualitative approach can be adapted in countries with similar usage patterns to develop effective communication strategies. This methodology can be used in low resource settings due to its simplicity and ease of implementation.

## Supporting information

**S1 File. Focus group guide.**
(DOC)

**S2 File. In-depth interview guide.**
(DOC)

**S3 File. COREQ check list.**
(PDF)

**S4 File. FCTC 2030 detail presentation.**
(PPT)

## Acknowledgments

Dr Achini Jayatilake, Dr Dilanka Delphachithra, the Medical Officers of Health and the Public Health Inspectors of each location of the study, all the study participants for their valuable

contribution to carry out the study. We also acknowledged the staff of the National Authority on Tobacco and Alcohol.

## Author Contributions

**Conceptualization:** Hemantha Amarasinghe, Palitha Abeyakoon.

**Data curation:** Hemantha Amarasinghe, Nilantha Ratnayake.

**Formal analysis:** Hemantha Amarasinghe, Nilantha Ratnayake, Sajeeva Ranaweera.

**Funding acquisition:** Thirupathy Suveendran, Palitha Abeyakoon.

**Investigation:** Nilantha Ratnayake.

**Methodology:** Hemantha Amarasinghe, Nilantha Ratnayake, Sajeeva Ranaweera.

**Project administration:** Sajeeva Ranaweera.

**Validation:** Sajeeva Ranaweera, Thirupathy Suveendran.

**Writing – original draft:** Hemantha Amarasinghe.

**Writing – review & editing:** Hemantha Amarasinghe, Sajeeva Ranaweera, Thirupathy Suveendran, Palitha Abeyakoon.

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
