## [Decision Letter · Decision Letter 0]

7 Jul 2022

PGPH-D-22-00450

Qualitative analysis to Identify determinants of use among different occupational settings and channels of communication to address smokeless tobacco use in Sri Lanka

Dear Dr. Amarasinghe,

Thank you for submitting your manuscript to PLOS Global Public Health. After careful consideration, we feel that it has merit but does not fully meet PLOS Global Public Health’s publication criteria as it currently stands. Therefore, we invite you to submit a revised version of the manuscript that addresses the points raised during the review process.

Please address all the comments raised by the reviewers. 

We look forward to receiving your revised manuscript.

Kind regards,

Palash Chandra Banik, MPhil

Academic Editor

Journal Requirements:

1.  Please amend your detailed online Financial Disclosure statement. This is published with the article. It must therefore be completed in full sentences and contain the exact wording you wish to be published.

2. Please update your online Competing Interests statement. If you have no competing interests to declare, please state: “The authors have declared that no competing interests exist.”

3. In the online submission form, you indicated that “Field visit notes and script of audio tape recordings are available on request from the corresponding author.”. All PLOS journals now require all data underlying the findings described in their manuscript to be freely available to other researchers, either 1. In a public repository, 2. Within the manuscript itself, or 3. Uploaded as supplementary information.

4. We have noticed that you have uploaded Supporting Information files, but you have not included a list of legends. Please add a full list of legends for your Supporting Information files after the references list. 

Additional Editor Comments (if provided):

Reviewers' comments:

Reviewer's Responses to Questions

**Comments to the Author**

1. Does this manuscript meet PLOS Global Public Health’s publication criteria? Is the manuscript technically sound, and do the data support the conclusions? The manuscript must describe methodologically and ethically rigorous research with conclusions that are appropriately drawn based on the data presented.

Reviewer #1: Yes

Reviewer #2: Yes

2. Has the statistical analysis been performed appropriately and rigorously?

Reviewer #1: N/A

Reviewer #2: Yes

3. Have the authors made all data underlying the findings in their manuscript fully available (please refer to the Data Availability Statement at the start of the manuscript PDF file)?

Reviewer #1: Yes

Reviewer #2: Yes

4. Is the manuscript presented in an intelligible fashion and written in standard English?

Reviewer #1: Yes

Reviewer #2: Yes

5. Review Comments to the Author

Reviewer #1: The authors have completed a qualitative study examining the self-described reasons for smokeless tobacco (SLT) use in Sri Lanka. The methods used are typical of this study style but must be interpreted with caution give then lack of representativeness of the subjects. While the study is interesting as written, I have several concerns that should be addressed.

• My most important critique is that I don’t have a sense of the comprehensiveness of the community sampling. The 13 communities could constitute 10% or 90% of Sri Lankan society. Second, I also have no idea how big one is versus another. In the current personation, I would have to assume that these communities are the same size. If there are large differences in the size of these communities, then that knowledge would inform the potential reach of any interventions.

• The authors should take care to ensure that the paper can be understood by non-Sri Lankans. For example, the authors use a term ‘estate sector’ and I am not sure what that means. It seems to be a type of plantation farming, but this needs to be explained more fully.

• In the intro, the authors described tobacco use in Sri Lankan, but they don’t clear state the entirety of tobacco use in the country. Specifically, I was unable to assess whether there are forms of tobacco between cigarettes and SLT as seen in many societies, such as waterpipe or cigar. And is marijuana smoking common? If SLT and cigarettes cover most tobacco use, please clearly state the percentages that use each individually versus dual use. The data from refence 5 left this unclear.

• I am not familiar with the term ‘Purposive sampling.’

• The authors describe many different communities, but I found some of the labels unusual. In the results they sometime describe the communities as occupation, but that clearly doesn’t apply to ‘Muslim’ or ‘Urban young adults,’ I can see that they are all communities, so I would avoid the term occupation.

• Including the names of TV channels is not informative for an international audience.

• The authors should be careful to use the same names in each instance. In the table we learn about “Urban young adults” but in the text there are also “Urban youth.” Are these the same people?

• The last sentence of the first paragraph of the discussion is an empty statement that doesn’t convey something one wouldn’t have guessed in advance: “These findings clearly demonstrate that when communications on preventing smokeless tobacco is planned, it needs to be carried out after careful consideration of many factors to obtain effective outcomes.”

• On page 19, an intervention is described as very costly, but there is consistent comparison of costs for different interventions, and this is an important factor. If the most effective message is costly, it still may be cost effective. In many societies, media is required to carry Public Service Announcements, and this may mitigate some costs. Since this not a cost effectiveness paper, I suggest you minimize discussion of costs.

• On page 20, the authors write “approached effectively by religious leaders.” Although suggested by participants, the authors have not gauged the effectiveness of any of these approaches.

• On page 20, the authors suggest there may not be an effective response to claims of the utility of SLT for “to address loneliness or to prevent a bad odor.” I’m sure there are alternative to SLT for these issues, so I don’t think this should be presented as hopeless.

Reviewer #2: The manuscript is technically sound and the data does support the conclusion made in the manuscript. The manuscript uses qualitative analysis and applied thematic analysis across occupational groups.

However there are few questions for the authors:

1. There is no framework given of how the number of interviews and FDGs were finalized. Was it based on some methodology?

2. First paragraph of the results section states "Females were represented among estate workers, garbage collectors of the local civil administration and Muslim communities and income remain constant for each occupation". However, no comparison is made by gender in the thematic analysis?

3. More emphasis can be done on the factors which are overlapping in different occupations group. As it is easy for the regulatory body to implement interventions which are common to many occupational groups. After which specific reforms can be introduced for each group.

6. PLOS authors have the option to publish the peer review history of their article (what does this mean?). If published, this will include your full peer review and any attached files.

**Do you want your identity to be public for this peer review?** For information about this choice, including consent withdrawal, please see our Privacy Policy.

Reviewer #1: No

Reviewer #2: No

---

## [Decision Letter · Decision Letter 1]

11 Nov 2022

Qualitative analysis to Identify determinants of use among different occupational settings and channels of communication to address smokeless tobacco use in Sri Lanka

PGPH-D-22-00450R1

Dear Dr. Amarasinghe,

We are pleased to inform you that your manuscript 'Qualitative analysis to Identify determinants of use among different occupational settings and channels of communication to address smokeless tobacco use in Sri Lanka' has been provisionally accepted for publication in PLOS Global Public Health.

Best regards,

Palash Chandra Banik, MPhil

Academic Editor

Reviewer Comments (if any, and for reference):

Reviewer's Responses to Questions

**Comments to the Author**

1. If the authors have adequately addressed your comments raised in a previous round of review and you feel that this manuscript is now acceptable for publication, you may indicate that here to bypass the “Comments to the Author” section, enter your conflict of interest statement in the “Confidential to Editor” section, and submit your "Accept" recommendation.

Reviewer #1: All comments have been addressed

2. Does this manuscript meet PLOS Global Public Health’s publication criteria? Is the manuscript technically sound, and do the data support the conclusions? The manuscript must describe methodologically and ethically rigorous research with conclusions that are appropriately drawn based on the data presented.

Reviewer #1: Yes

3. Has the statistical analysis been performed appropriately and rigorously?

Reviewer #1: Yes

4. Have the authors made all data underlying the findings in their manuscript fully available (please refer to the Data Availability Statement at the start of the manuscript PDF file)?

Reviewer #1: (No Response)

5. Is the manuscript presented in an intelligible fashion and written in standard English?

Reviewer #1: Yes

6. Review Comments to the Author

Reviewer #1: The authors have addressed my comments and made edits as needed.

7. PLOS authors have the option to publish the peer review history of their article (what does this mean?). If published, this will include your full peer review and any attached files.

**Do you want your identity to be public for this peer review?** For information about this choice, including consent withdrawal, please see our Privacy Policy.

Reviewer #1: No
